# Eating Disorders in an Immigrant Population: Are Clinical Features and Treatment Outcomes Different from the Native-Born Spanish Population?

**DOI:** 10.3390/nu17243914

**Published:** 2025-12-14

**Authors:** Magda Rosinska, Silvia Tempia Valenta, Isabel Sánchez, Olga Jordana Ovejero, Teresa Alonzo-Castillo, Laura Gálvez Solé, Rosa Fontana Eito, Lucero Munguia, Elena Caravaca Sanz, Anna Rita Atti, Roser Granero, Susana Jiménez-Murcia, Fernando Fernández-Aranda

**Affiliations:** 1Clinical Psychology Department, Bellvitge University Hospital, 08907 Barcelona, Spainsjimenez@bellvitgehospital.cat (S.J.-M.); 2Psychoneurobiology of Eating and Addictive Behaviors Group, Neuroscience Program, Bellvitge Biomedical Research Institute (IDIBELL), 08908 Barcelona, Spain; roser.granero@uab.cat; 3CIBER Physiopathology of Obesity and Nutrition (CIBERobn), Instituto Salud Carlos III, 08907 Barcelona, Spain; 4Department of Biomedical and Neuromotor Sciences, University of Bologna, 40126 Bologna, Italy; 5Research Group on Emergency and Disaster Medicine, Vrije Universiteit Brussel, 1050 Brussels, Belgium; 6Department of Clinical Sciences, School of Medicine and Health Sciences, University of Barcelona, 08907 Barcelona, Spain; 7Department of Psychiatry, Vall d’Hebron University Hospital, 08035 Barcelona, Spain; 8Department of Psychobiology and Methodology of Health Sciences, Autonomous University of Barcelona, 08193 Barcelona, Spain

**Keywords:** eating disorders, personality traits, immigrant, native-born, acculturation

## Abstract

**Background/Objectives:** Sociocultural factors, including migration and acculturation, may influence the clinical profile and course of eating disorders (EDs). This study examined differences between immigrant and native-born Spanish patients with EDs in (1) clinical presentation and (2) treatment response. **Methods:** Consecutive outpatients from the Eating Disorders Unit at Bellvitge University Hospital (Barcelona, Spain) were assessed using the Eating Disorder Inventory-2 (EDI-2), Symptom Checklist-90-R (SCL-90-R), and Temperament and Character Inventory-Revised (TCI-R). Statistical analyses included chi-square tests, ANOVA, Cox regression for dropout, and logistic regression for predictors of poor outcome, adjusted for ED subtype. **Results:** The sample included 1104 patients (947 native-born; 157 immigrants). Immigrant patients showed a distinct clinical profile, with lower drive for thinness and body dissatisfaction but higher interpersonal distrust, maturity fears, perfectionism, anxiety-related symptoms, and self-transcendence. They also presented a worse treatment response, including higher dropout rates, poorer outcomes, and lower remission rates. Predictive models identified different risk factors for poor treatment response in each group: among native-born patients, younger age of ED onset, higher novelty seeking, and lower self-directedness were associated with worse outcomes, whereas among immigrant patients, greater ED severity, lower harm avoidance, and lower self-transcendence predicted poorer results. **Conclusions:** Immigrant patients with EDs exhibit a differentiated clinical presentation and less favorable treatment response compared to native-born patients. The differential predictors of poor outcome highlight the need for culturally informed and individually tailored interventions that consider both sociocultural context and personality-related vulnerabilities.

## 1. Introduction

Eating disorders (EDs) are severe mental illnesses that can affect any individual [1,2]. The onset and maintenance of EDs are influenced by various factors, including personality traits and psychosocial variables [1]. Common factors across EDs include concerns about weight and shape, anxiety, depression, and certain personality traits, such as harm avoidance and low self-directedness [3,4]. While these factors tend to fluctuate across ED subtypes, such as anorexia nervosa (AN), bulimia nervosa (BN), binge eating disorder (BED), and other specified feeding and eating disorders (OSFED) [5], they play a significant role in treatment responses, including relapse, dropout, treatment completion [6], and seeking or referral to treatment [7].

Beyond individual psychological factors, broader sociocultural variables, such as migration, have been examined for their role in ED risk [8,9]. Studies suggest that migration can act as either a protective or risk factor for EDs, depending on demographic variables and individual experiences [10,11]. However, results are mixed. Some studies report elevated risks for EDs and body dissatisfaction in immigrant populations, such as Mexican immigrants in the USA compared to those in Mexico, Zimbabwean women in the UK compared to those in Zimbabwe, and North African immigrant women compared to their counterparts in North Africa [12,13,14]. Other studies show lower ED rates in populations like first-generation immigrants in Australia compared to the host population, immigrant women in Israel compared to local-born Israelis, and immigrants in Finland compared to native Finns [15,16,17]. In Spain, two large studies conducted over a decade ago found higher ED risks among immigrant women living in Spain for fewer than six years, as well as greater ED risk in immigrant adolescents compared to their Spanish peers [18,19]. More recently, a large meta-analysis confirmed that disordered eating is associated with immigrant status [20].

Acculturation, the process of adjusting to and balancing one’s culture of origin with that of a new country, has also been identified as an important factor [21]. This process often involves adapting to a new language, values, social practices of the dominant society in a new cultural environment, which can result in increased distress [22]. Sociocultural factors are particularly influential in shaping body image, desired weight, and self-perception, especially when immigrants adopt Western values, including the cultural ideal of thinness and associated eating behaviors [23,24]. It is plausible that cultural shifts, particularly exposure to Western norms, and the process of acculturation may contribute to ED vulnerability among immigrant populations [25,26]. In terms of personality traits, research suggests that traits like conscientiousness, emotional stability, and extraversion may mediate the relationship between ethnicity and ED, particularly in aspects like body esteem [27].

Given the heterogeneity of immigrant populations and the limited research in this area, this study aimed to examine whether immigrant and native-born patients with EDs differ in their (1) clinical presentation and (2) treatment response. Specifically, we explored group differences in ED symptoms, associated psychopathology, personality traits, and outcomes. We hypothesized that (1) immigrant patients would show a distinct clinical and psychological profile compared to native-born patients, and that (2) they would have higher dropout rates and poorer treatment outcomes.

## 2. Materials and Methods

### 2.1. Participants and Procedure

This sample consisted of 1104 participants, including 947 native-born individuals and 157 immigrants. The participants were consecutively recruited from the ED Unit at Bellvitge University Hospital in Barcelona, Spain. All participants met the diagnostic criteria outlined in the Diagnostic and Statistical Manual of Mental Disorders, 5th edition (DSM-5) [28], by means of a semi-structured face-to-face interview.

### 2.2. Treatment

For patients with AN, treatment addressed both nutritional patterns and psychological/psychiatric aspects, based on a proven cognitive behavioral therapy (CBT) program [29,30]. Patients attended the hospital daily from 9:00 AM to 3:00 PM, Monday through Friday, for a 16-week period. Two daily meals, breakfast and lunch, were monitored during this time.

For patients with BN, BED, and OSFED, treatment consisted of weekly 90-min therapy sessions over a 16-week period, led by experienced psychologists. This treatment program, published in Spanish, has demonstrated efficacy [31,32]. The goals of the therapy included education on problem-solving strategies, cognitive restructuring, emotional regulation, and improving self-esteem and body image, as well as relapse prevention. Additionally, the therapy addressed eating-related symptomatology through psychoeducation, dietary monitoring, and normalization of nutritional patterns.

### 2.3. Assessment

#### 2.3.1. Semi-Structured Clinical Interview

Sociodemographic information including sex, civil status, education, employment, socioeconomic status, clinically relevant features regarding ED and psychopathological symptoms was assessed by a structured clinical interview.

#### 2.3.2. Self-Report Measures

Each participant completed a series of validated self-report questionnaires evaluating psychopathological symptoms and temperament traits. Cronbach’s alpha values were reported for the questionnaires assessed; Cronbach’s alpha measures internal consistency, and values ≥ 0.70 are generally considered indicative of acceptable reliability in research studies [33].

Eating Disorder Inventory-2 (EDI-2) [34] is a self-report questionnaire made up of 91 questions looking at psychological and behavioral characteristics in relation to EDs on a 6-point Likert scale. There are 11 subscales: drive for thinness, body dissatisfaction, bulimia, ineffectiveness, perfectionism, interpersonal distrust, interoceptive awareness, maturity fears, asceticism, impulse regulation, and social insecurity. It has been validated in the general population in Spain [35]. In this study, Cronbach’s alpha for the EDI-2 ranged from 0.718 (asceticism) to 0.953 (total scale).

Symptom Checklist-90-Revised (SCL-90-R) [36] consists of 90 items and is a self-report questionnaire designed to assess distress and psychopathology. The questionnaire includes nine primary dimensions: somatization, obsessive-compulsive behavior, interpersonal sensitivity, depression, anxiety, hostility, phobic anxiety, paranoid ideation, and psychoticism. Additionally, it provides three global indices: the Global Severity Index (GSI), the Positive Symptom Total (PST), and the Positive Symptom Distress Index (PSDI). It has been validated in the general Spanish adult population [37]. In this study, Cronbach’s alpha for the SCL-90-R ranged from 0.758 (paranoia) to 0.977 (global indices).

Temperament and Character Inventory-Revised (TCI-R) [38] is a self-report questionnaire consisting of 240 items, scored on a 5-point Likert scale. The questionnaire includes four temperament scales: novelty seeking, harm avoidance, reward dependence, and persistence, as well as three character scales: cooperativeness, self-directedness, and self-transcendence. Also, this scale has been validated in the general adult population in Spain [39]. In the current sample Cronbach’s alpha ranged from α = 0.806 (novelty seeking) to α = 0.905 (harm avoidance & persistence).

### 2.4. Procedure and Ethics

All participants voluntarily sought treatment and provided consent to participate in this study. They were all diagnosed with an ED in line with the DSM-5 criteria [28] by clinical psychologists and psychiatrists with over 15 years’ experience in the ED field. Clinical interviews were conducted face-to-face prior to and after psychometric evaluations. These evaluations were completed via the self-report questionnaires with pencil and paper. During the first visit, a clinical interview was conducted, followed by provision of test results and treatment details during subsequent visits, based on each patient’s clinical profile. No compensation for this study was provided to the patients.

According to the Declaration of Helsinki, the present study was approved by the Clinical Research Ethics Committee (CEIC) of the University Hospital of Bellvitge. Additionally, written and informed consent was obtained from all participants in this study.

### 2.5. Statistical Analysis

SPSS 29 (IBM Corp., Armonk, NY, USA) was employed for the statistical analysis. The sociodemographic profile was compared between the groups with chi-square tests (χ^2^) for categorical variables and with analysis of variance (ANOVA) for quantitative variables (since this study includes only two groups, the ANOVA results are identical to those obtained using an independent-samples Student’s *t*-test for comparing means).

To assess differences between the groups at baseline for the ED symptom level (EDI-2 scales), general psychopathology (SCL-90-R scales) and personality traits (TCI-R scales), ANOVA and post-hoc tests were employed. Effect sizes were reported using Cramer’s V (for categorical comparisons) and Cohen’s d (for quantitative comparisons). For Cramer’s V, values of approximately 0.10, 0.30, and 0.50 were considered small, medium, and large effects, respectively. For Cohen’s d, values around 0.20, 0.50, and 0.80 were interpreted as small, medium, and large effect sizes.

The risk of treatment outcomes (dropout and remission status) was calculated, and comparison between groups adjusted for ED subtypes yielded significant results. A predictive model aimed at identifying the variables related to poor treatment outcome was also obtained through stepwise logistic regression, defining as predictors the age of onset of the ED, ED severity level (as measured with the EDI-2 total score), the personality dimensions (TCI-R scores). Separate models were obtained, stratified by immigrant status, to evaluate the potential role of this feature as a moderation/interaction term. The goodness-of-fit of the selected predictive models was tested using the Hosmer–Lemeshow test (*p* > 0.05 indicates adequate fitting) and the global predictive capacity was measured with Nagelkerke’s pseudo R^2^.

Finally, the cumulative survival curves for the dropout rate during treatment were estimated with Cox’s regression, including the ED subtype as a covariate.

In this work, the increase in the Type-I error due to the employment of multiple significance tests was controlled with Finner’s procedure.

## 3. Results

### 3.1. Description of the Sample

The majority of immigrant patients were from Latin America (71.34%), followed by Europe (Eastern: 6.37%, Other: 13.38%), Africa (8.28%), and Asia (0.64%) (Appendix A). The majority of participants identified as female, with no significant gender differences observed between the two groups. Similarly, no significant differences were observed in civil status, educational levels, and social position indexes between the two groups. A higher percentage of immigrant participants were unemployed (42.7%) compared to native-born patients (34.2%).

Regarding clinical characteristics, no differences between groups were identified for the means of chronological age, onset and duration of the disordered eating. There were differences in the distribution of ED subtypes between the groups, with a higher proportion of AN and BED among native-born participants, and higher proportion of BN and OSFED among immigrant individuals. Table 1 shows the detailed description of the sample.

### 3.2. Differences in the Baseline Measures Between Immigrant and Native-Born Patients

The result of the ANOVA comparing the measures registered at pre-treatment is displayed in Table 2, and the visualization of the profiles is plotted in the radar-charts displayed in Figure 1 and the bar-chart displayed in Figure 2 (this figure shows the percentage of patients with standardized T-scores above 63, which on the SCL-90-R are considered significantly elevated and indicate that these individuals exhibit symptoms that deviate from the population average and may have potential clinical relevance). These results revealed differences between the groups in all the domains (eating psychopathology, psychopathology, and personality).

Concretely, the EDI-2 evidenced that immigrant patients scored lower means on drive for thinness and body dissatisfaction, and higher means on interpersonal distrust, maturity fears and perfectionism. Marginally significant results indicated that immigrant patients also scored higher on interoceptive awareness and impulse regulation. Regarding the psychopathology state, immigrant patients reported higher levels of anxiety and phobic anxiety. Considering the personality traits, immigrant patients scored higher on persistence and self-transcendence and lower on reward dependence and harm avoidance.

### 3.3. Differences in Treatment Outcomes Between Immigrant and Native-Born Patients

The risk of treatment outcomes is displayed in Table 3 and Figure 3. Post-hoc comparisons for the outcome analyzed in four groups (dropout, non remission, partial remission and full remission) showed that compared to native-born patients, immigrant patients were more likely to dropout (*p* = 0.003). Considering the binary outcome poor versus good outcome, immigrant patients also achieved higher likelihood of poorer results. These differences remained significant even after adjusting for ED subtypes (the contrast for comparing the likelihood of dropout between native-born and immigrant patients achieved significant result *p* = 0.014).

The two logistic regressions selecting the best predictors for poor treatment outcomes retained distinct factors among immigrant and native-born patients (Table 4). Adequate fitting indexes were achieved (*p* > 0.05 in the Hosmer-Lemeshow tests). Within the native-born group, younger age of onset of ED, higher novelty seeking, and lower self-directedness were associated with the higher likelihood of poorer outcome. Among immigrant participants, higher overall ED symptom severity, lower harm avoidance and lower self-transcendence predicted poorer outcomes. The regression model obtained for immigrant patients in Table 4 showed a higher Nagelkerke’s pseudo-R^2^ coefficient compared to the model for native-born individuals, despite both models including the same number of predictors. This difference indicates better overall predictive performance in Model 2.

### 3.4. Survival Curves for the Rate of Treatment Dropout

The cumulative survival curves for treatment dropout are presented in Figure 4, with plots generated using Cox regression adjusted for ED subtype. The analyses yielded significant results, indicating higher dropout rates among immigrant patients. Concretely, the survival functions showed similar early retention but a higher cumulative dropout rate over time among immigrant patients.

## 4. Discussion

Our study included 947 native-born patients and 157 immigrant patients, with immigrants comprising approximately 14.2% of the sample, a percentage closely mirroring Spain’s foreign-born population, which, as of 2024, accounted for 18.78% of the total population [40]. We observed significant differences in the distribution of ED subtypes between the two groups. These differences likely reflect real patterns within our clinical catchment area, where cultural norms, help-seeking behaviors, symptom expression, and access to specialized care vary across populations; however, because we did not collect migration-specific or cultural variables, we cannot determine the precise cultural or contextual factors driving these subtype disparities. Beyond subtype distribution, our findings revealed that immigrant patients exhibited less concern with thinness and body image but significantly greater interpersonal distrust, maturity fears, and perfectionism. They also showed higher levels of anxiety and phobic anxiety, as well as distinct personality profiles characterized by lower harm avoidance and reward dependence but higher persistence and self-transcendence. In terms of treatment outcomes, immigrant patients were more likely to drop out, achieve poorer results, and have lower remission rates compared to native-born patients. Predictive models identified different risk factors for poor therapy response in native-born and immigrant patients. For native-born patients, younger age at ED onset, higher novelty seeking, and lower self-directedness were predictors of worse outcomes. For immigrant patients, greater ED severity, and lower harm avoidance and self-transcendence were linked to poorer treatment outcomes. While we frame some of the observed psychopathological differences within theoretical models of acculturative stress and cultural adjustment, it is important to note that we did not directly measure acculturation, migration-related stress, or cultural conflict; therefore, these interpretations should be considered exploratory and interpreted with caution.

The clinical features of EDs are influenced by cultural practices and ethnic identity. Historically EDs have been considered Western disorders, partly due to the pervasive thinness ideal promoted by the media since the mid-20th century [41,42]. Indeed, it has been demonstrated that the incidence of EDs tends to increase in populations migrating to Western countries [41,42]. This phenomenon highlights how individuals may absorb preconceived notions about physical appearance and eating habits through the process of acculturation [26,43]. It has also been shown that eating symptomatology could improve depending on the individual’s connection to their local-born culture: literature shows how ethnic identity can act as a protective factor against dysfunctional eating behaviors [26].

Our study supports this perspective, showing that the immigrant population was less concerned about thinness and physical appearance compared to the native-born Spanish population. However, they showed higher levels of interpersonal distrust, maturity fears, perfectionism, interoceptive awareness and impulse regulation, which could potentially reflect cultural conflict-related distress. Interpersonal distrust and fears about maturity may plausibly relate to challenges in building relationship and in handling difficulties within the host culture, potentially influenced by experiences of marginalization or stigma [44]. Higher perfectionism could reflect maladaptive self-criticism or internalized pressures to meet the perceived standards of the host culture [45]. Elevated interoceptive awareness may point to an increased focus on bodily and emotional states under stress, a pattern described in prior studies on cultural conflict and emotional regulation [11]. Impulse regulation, although generally adaptive, might reflect in this context an excessive effort to conform to or suppress emotional distress [46]. Previous research has also shown that discrepancies between the host culture’s aesthetic norms and those of an individual’s cultural background can intensify eating-related symptoms, particularly in individuals strongly attached to their cultural heritage [47].

The results regarding general psychopathology and personality traits show a distinct psychological profile between immigrant and native patients. The higher levels of anxiety-related symptoms, paranoia, and psychotic symptoms suggest that individuals who have experienced a migration history, as widely demonstrated in the literature, tend to face greater psychological distress often linked to socioeconomic difficulties and challenges of integration [48,49]. The elevated paranoia and psychotic symptoms may stem from traumatic factors or reflect feelings of social alienation, loneliness or mistrust within the host society [50]. In terms of personality traits, our findings show lower levels of harm avoidance and reward dependence, which seem to indicate a greater propensity for potentially harmful situations and a lower reliance on approval or social reinforcement [51]. This could suggest a more independent coping style and appears consistent with the interpersonal distrust observed. Conversely, higher persistence and self-transcendence might indicate a determination to overcome challenges and a strong focus on meaning or connection beyond oneself, aspects that seem predominantly cultural and could serve as protective factors against some of the adversities faced [52,53].

To date, to our knowledge, no studies have specifically examined ED treatment outcomes in immigrant populations. However, insights can be drawn from research on psychological treatments for immigrant populations in other mental health disorders. Recent literature found that immigration status can significantly impact the severity of somatoform and depressive symptoms at admission, with immigrant patients often presenting with greater symptom severity [54] and more previous stressful situations. While psychotherapy effectively reduced depressive and anxious symptoms, somatization and posttraumatic stress symptoms showed less improvement [54]. For refugees with post-traumatic stress disorder (PTSD), CBT has been shown to demonstrate large effect sizes, supporting its broad suitability for treating core PTSD symptoms [55]. A recent meta-analysis showed that dropout rates in refugee populations are comparable to those in non-refugee populations [56]. In contrast, our findings revealed higher dropout rates, poorer outcomes, and lower remission rates in immigrant patients compared to native-born patients: in the context of EDs, the ego-syntonic nature of these disorders, coupled with cultural and migratory stressors, may create a cycle of non-compliance and treatment resistance [57,58]. Greater economic hardship and family priorities, often for first-degree relatives who have remained in the country of origin, add to the stress and the need to prioritize work over treatment. Predictors such as lower harm avoidance and self-transcendence, which in our study were linked to worse outcomes, likely reflect deeper psychological and cultural factors that will require further exploration. Lower harm avoidance could indicate a lower tendency to evade risks or potentially harmful situations, perhaps reflecting a coping style shaped by prior adversity or cultural values [59]. Similarly, self-transcendence, often linked to a focus on meaning and connection beyond the self, may interact with cultural or individual factors in ways that influence engagement in treatment and recovery [60,61].

## 5. Limitations and Strengths

The limitations of our work are several and should be considered when interpreting the findings. First, we lacked detailed information on the migration process, including reasons for migration, migration-related stressors, and potential traumatic experiences occurring before or after migration. We also did not collect key variables such as legal status, migrant generation, level of acculturation, or possible language barriers during treatment. These factors are known to influence ED onset, clinical course, and treatment response. While we did assess core sociodemographic characteristics (i.e., sex, civil status, education, employment, and socioeconomic status) through a structured clinical interview, this information cannot replace the migration-specific variables that would have allowed for a more nuanced interpretation of the differences between native-born and immigrant patients. Second, we did not examine cultural variables such as race or ethnicity, nor culturally shaped eating practices (e.g., family meal traditions, religious or cultural celebrations involving food). Likewise, we did not evaluate potential cultural variation in personality constructs. Since personality measures such as the TCI-R are not culturally neutral, the lack of population-specific norms for immigrant patients represents an additional source of measurement bias that must be acknowledged. Furthermore, by grouping all immigrants into a single category, we inevitably overlook relevant cultural heterogeneity in body image ideals, eating-related norms, and illness expression. Given the small size of the migrant sample, subgroup analyses were not feasible, but this simplification must be recognized as an additional limitation affecting interpretation of cross-group differences. Third, we did not systematically assess linguistic or communicative differences. Although patients admitted to treatment were required to have sufficient language comprehension to participate, we did not quantify language proficiency nor examine whether communication style differences, subtle language barriers, or culturally shaped emotional expression may have influenced treatment engagement or outcomes. Given that ED treatment relies heavily on verbal processing and emotional exploration, this omission is an important limitation. In this regard, although all participants met the required language criterion and received the same standardized CBT protocol, we acknowledge that immigrant patients may still experience subtler challenges in adapting to an intervention originally developed for the native population. Importantly, one of the aims of this study is precisely to gather the empirical evidence needed to understand the specific needs of immigrant patients so that future adaptations of standard treatments can be informed and implemented appropriately. Fourth, although no differences emerged in education or social index, we observed significantly lower employment rates among immigrants, which may reflect broader socioeconomic adversities. These contextual factors may shape how EDs develop, manifest, and respond to treatment. Fifth, we were unable to distinguish between first- and second-generation immigrants due to insufficient data, preventing exploration of potentially important intergenerational differences. Sixth, there was a marked imbalance between native-born and immigrant patients, as well as between women and men. These discrepancies reflect the actual distribution of patients in our clinical setting, but they nonetheless limit generalizability and reduce statistical power for subgroup analyses. Finally, this was a single-center study, and the characteristics of the region and the population treated in our unit must be considered. Consequently, the findings cannot be generalized beyond similar clinical settings.

However, this study had some notable strengths such as being a novel study in terms of looking at treatment outcome in a large clinical sample of immigrant and native-born patients with EDs, as well as comparing eating and psychopathological symptoms and personality traits of the two groups. Future research should focus on specific races, cultures, and migration pathways to obtain more accurate and culturally sensitive data on different populations. It is in fact desirable that future studies provide guidance to design new treatments specifically tailored to different immigrant populations.

## 6. Conclusions

In conclusion, the present study showed significant differences in clinical characteristics, personality traits and treatment outcomes between immigrant and native-born Spanish patients with EDs. Immigrant participants showed higher levels of interpersonal difficulties, perfectionism and cultural conflict-related distress, which were associated with poorer treatment outcomes and higher dropout rates. These findings show how the psychopathological picture can undergo strong cultural influences, exhibiting a changing and ambiguous symptomatology. Both estrangement from one’s cultural heritage and the clash with new customs and aesthetic standards appear to contribute to the clinical picture. It is notable that treatments designed for the local population tend to show reduced efficacy in the immigrant populations. It is necessary today, in light of an increasingly diverse and multicultural society, to develop culturally sensitive and effective treatments tailored to the needs of each individual.

## Figures and Tables

**Figure 1 nutrients-17-03914-f001:**
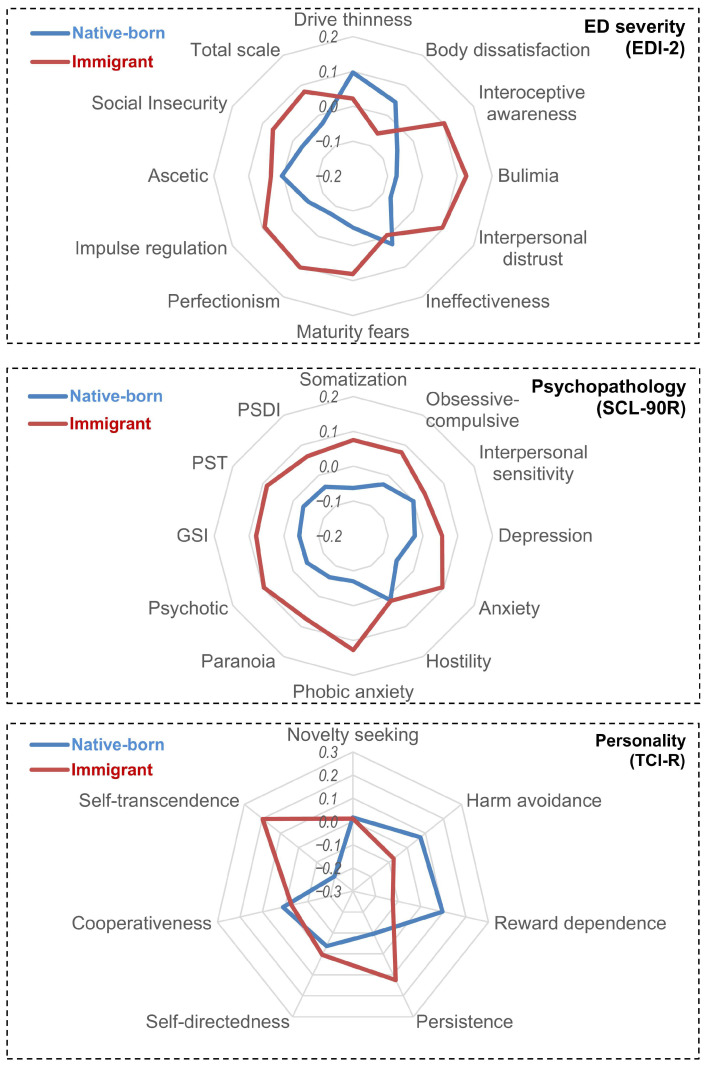
Radar Plot Comparing Eating Psychopathology, General Psychopathology, and Personality Traits in Native-Born vs. Immigrant Patients at Baseline.

**Figure 2 nutrients-17-03914-f002:**
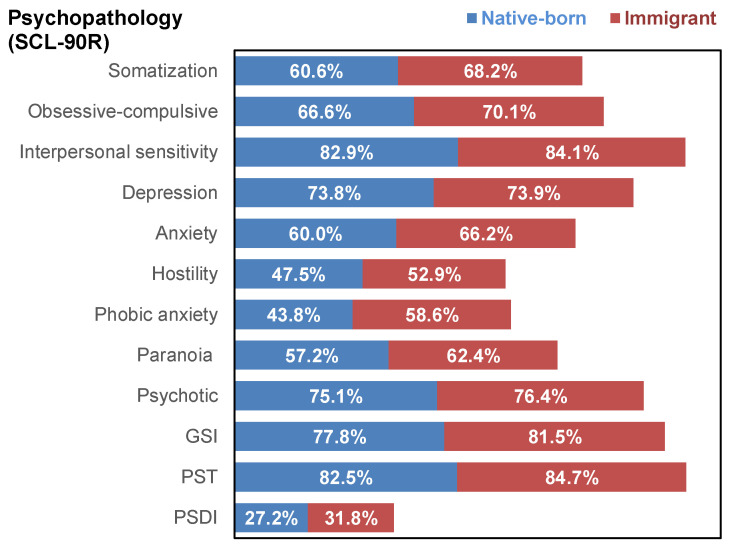
Percentage of Patients Scoring Within the Clinical Range (T-scores higher than 63 compared with the standardization sample) Across Major Psychopathology Domains.

**Figure 3 nutrients-17-03914-f003:**
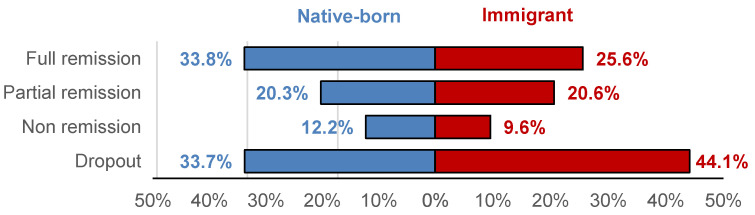
Treatment Outcomes in Native-Born vs. Immigrant Patients (Adjusted for ED Subtype). Note. Risk estimates adjusted to ED subtype.

**Figure 4 nutrients-17-03914-f004:**
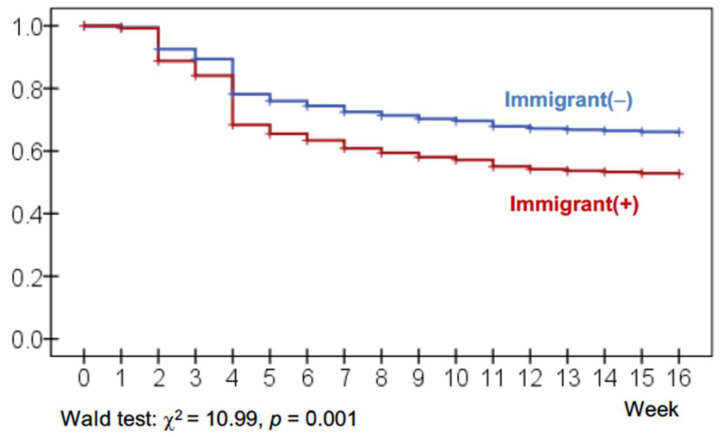
Cumulative Survival Curves for Treatment Dropout. Note. Cox’s regression adjusted to ED subtype.

**Table 1 nutrients-17-03914-t001:** Sociodemographic and Clinical Characteristics of Native-Born and Immigrant Patients at Baseline.

	Native-Born (*n* = 947)		Immigrant (*n* = 157)			
	** *N* **	**%**	** *n* **	**%**	** *p* **	**C-V**
Sex						
Women	864	91.2%	149	94.9%	0.122	0.047
Men	83	8.8%	8	5.1%		
Civil status						
Single	662	69.9%	117	74.5%	0.495	0.036
Married	203	21.4%	29	18.5%		
Divorced	82	8.7%	11	7.0%		
Education						
Primary	337	35.6%	50	31.8%	0.628	0.029
Secondary	424	44.8%	76	48.4%		
University	186	19.6%	31	19.7%		
Employment						
Unemployed	324	34.2%	67	42.7%	**0.040 ***	0.062
Employed or student	623	65.8%	90	57.3%		
Social index						
High	11	1.2%	4	2.5%	0.296	0.067
Mean-high	61	6.4%	14	8.9%		
Mean	101	10.7%	19	12.1%		
Mean-low	271	28.6%	48	30.6%		
Low	503	53.1%	72	45.9%		
	**Mean**	**SD**	**Mean**	**SD**	** *p* **	**|d|**
Age (yrs)	29.38	10.91	27.68	9.47	0.065	0.17
Age of onset of ED (yrs)	20.55	9.23	19.11	8.22	0.067	0.16
Duration of ED (yrs)	8.99	8.32	8.66	6.93	0.642	0.04
ED subtype	** *n* **	**%**	** *n* **	**%**	** *p* **	**C-V**
AN	258	27.60%	21	13.40%	**<****0.001** *****	0.137
BN	287	30.70%	67	42.70%		
BED	166	17.80%	21	13.40%		
OSFED	224	24%	48	30.60%		

Note. AN: anorexia nervosa. BN: bulimia nervosa. BED: binge eating disorder. C-V: Cramer’s V (values of approximately 0.10, 0.30, and 0.50 indicate small, medium, and large effect sizes, respectively). |d|: absolute value of Cohen’s d (values around 0.20, 0.50, and 0.80 indicate small, medium, and large effect sizes, respectively). ED: eating disorder. OSFED: other specified feeding and eating disorder. SD: standard deviation. * Bold: significant comparison (0.05 level). *p*-values obtained applying Finner’s correction.

**Table 2 nutrients-17-03914-t002:** Baseline Differences in Eating Psychopathology, General Psychopathology, and Personality Traits (ANOVA Results).

	Native-Born (*n* = 947)		Immigrant (*n* = 157)			
	Mean	SD	Mean	SD	*p*	|d|
EDI-2 Drive thinness	13.93	5.69	12.52	5.46	**0** **.004 ***	0.25
EDI-2 Body dissatisfaction	17.27	7.57	15.50	7.27	**0** **.006 ***	0.24
EDI-2 Interoceptive awareness	11.15	6.91	12.23	6.63	0.066	0.16
EDI-2 Bulimia	6.82	4.42	7.39	4.25	0.132	0.13
EDI-2 Interpersonal distrust	5.43	4.93	6.63	4.73	**0** **.004 ***	0.25
EDI-2 Ineffectiveness	10.95	7.38	10.30	7.09	0.304	0.09
EDI-2 Maturity fears	7.80	5.74	8.79	5.51	**0** **.041 ***	0.18
EDI-2 Perfectionism	5.53	4.40	6.57	4.22	**0** **.005 ***	0.24
EDI-2 Impulse regulation	6.29	6.05	7.23	5.81	0.067	0.16
EDI-2 Ascetic	7.02	4.15	6.96	3.98	0.868	0.01
EDI-2 Social Insecurity	7.36	5.15	7.89	4.95	0.225	0.11
EDI-2 Total scale	99.55	42.40	102.04	40.72	0.490	0.06
SCL-90-R Somatization	1.75	0.93	1.89	0.89	0.078	0.15
SCL-90-R Obsessive-compulsive	1.85	0.88	1.93	0.84	0.278	0.09
SCL-90-R Interpersonal sensitivity	2.02	0.94	2.02	0.91	0.999	0.00
SCL-90-R Depression	2.20	0.94	2.26	0.90	0.411	0.07
SCL-90-R Anxiety	1.62	0.93	1.78	0.89	**0** **.042 ***	0.18
SCL-90-R Hostility	1.37	1.00	1.34	0.96	0.713	0.03
SCL-90-R Phobic anxiety	0.97	0.91	1.18	0.88	**0** **.006 ***	0.24
SCL-90-R Paranoia	1.44	0.87	1.58	0.83	0.054	0.17
SCL-90-R Psychoticism	1.30	0.77	1.43	0.74	0.054	0.17
SCL-90-R GSI	1.71	0.76	1.80	0.73	0.151	0.12
SCL-90-R PST	63.15	17.69	65.04	16.99	0.208	0.11
SCL-90-R PSDI	2.32	0.58	2.38	0.56	0.250	0.10
TCI-R Novelty seeking	100.94	16.42	99.66	15.77	0.357	0.08
TCI-R Harm avoidance	118.87	20.28	114.12	19.47	**0** **.006 ***	0.24
TCI-R Reward dependence	102.58	16.06	97.43	15.42	**<** **0** **.001 ***	0.33
TCI-R Persistence	107.85	20.65	115.00	19.83	**<** **0** **.001 ***	0.35
TCI-R Self-directedness	117.22	21.08	120.19	20.25	0.097	0.14
TCI-R Cooperativeness	133.93	15.92	133.66	15.29	0.842	0.02
TCI-R Self-transcendence	64.20	15.48	72.93	14.87	**<** **0** **.001 ***	**0.58 ^†^**

Note. |d|: absolute value of Cohen’s d (values around 0.20, 0.50, and 0.80 indicate small, medium, and large effect sizes, respectively). ED: eating disorder. SD: standard deviation. * Bold: significant comparison (0.05 level). ^†^ Bold: relevant effect size (coefficient into the mild-moderate to large-high effect range). *p*-values obtained applying Finner’s correction.

**Table 3 nutrients-17-03914-t003:** Treatment Outcomes in Native-Born and Immigrant Patients, Before and After Adjustment for ED Subtype.

	Native-Born		Immigrant				Adjusted to ED Subtype		
	(*n* = 947)		(*n* = 157)				Native-Born	Immigrant	
	*n*	%	*n*	%	*p*	C-V	*%*	*%*	*p*
Dropout	323	34.1%	73	46.5%	**0** **.012 ***	0.10	33.7%	44.1%	**0** **.049 ***
Non remission	122	12.9%	16	10.2%			12.2%	9.6%	
Partial remission	202	21.3%	34	21.7%			20.3%	20.6%	
Full remission	300	31.7%	34	21.7%			33.8%	25.6%	
Poor outcome	445	47.0%	89	56.7%	**0** **.024 ***	0.07	46.0%	53.7%	0.072
Good outcome	502	53.0%	68	43.3%			54.0%	46.3%	

Note. C-V: Cramer’s V (values of approximately 0.10, 0.30, and 0.50 indicate small, medium, and large effect sizes, respectively). ED: eating disorder. * Bold: significant comparison (0.05 level).

**Table 4 nutrients-17-03914-t004:** Predictors of Poor Treatment Outcome by Immigrant Status (Logistic Regression Adjusted for ED Subtype).

**Native-Born;** ***n*** **= 947**	**B**	**SE**	** *p* **	**OR**	**95%CI OR**		**H-L**	**NR^2^**
Age of onset of ED (years)	−0.027	0.008	0.001	0.973	0.958	0.989	0.680	0.079
TCI-R Novelty seeking	0.009	0.004	0.043	1.009	1.000	1.018		
TCI-R Self directedness	−0.012	0.003	0.001	0.988	0.982	0.995		
**Immigrant;** ***n*** **= 157**	**B**	**SE**	** *p* **	**OR**	**95%CI OR**		**H-L**	**NR^2^**
EDI total score	0.013	0.005	0.013	1.013	1.003	1.024	0.669	0.136
TCI-R Harm avoidance	−0.023	0.012	0.047	0.977	0.954	1.000		
TCI-R Self transcendence	−0.030	0.013	0.018	0.970	0.946	0.995		

Note. B: unstandardized parameter. SE: standard error. OR: odds ratio. 95%CI: 95% confidence interval. H-L: Hosmer and Lemeshow test. NR^2^: Nagelkerke’s pseudo R^2^.

## Data Availability

All inquiries regarding availability of the data should be referred to the corresponding author (F.F.-A.), as there are ongoing studies using the data and to preserve patient confidentiality. Requests will be considered on a case-by-case basis.

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
