# Peer review of "Eating Disorders in an Immigrant Population: Are Clinical Features and Treatment Outcomes Different from the Native-Born Spanish Population?"

_nutrients, 2025, doi:10.3390/nu17243914_

Round 1

Reviewer 1 Report

Comments and Suggestions for Authors

Thank you for the opportunity to review the research article entitled ”Eating disorders in an immigrant population: are clinical features and treatment outcomes different from the native-born Spanish population?” It is a challenging research project, well-written, with potential for publication. 

In my opinion, the manuscript should be simplified and the research must be clarified, due to the many goals, variables, and hypotheses that the authors are presenting, even from the beginning: the research teams specified that there are results related to ... and please check the Abstract and also the manuscript:

”drive for thinness and body dissatisfaction .... interpersonal distrust, maturity fears, perfectionism, anxiety symptoms, and self-transcendence. Also, they present data about  ”dropout rates, poorer clinical outcomes, and lower remission...” Simultaneously, the title is focusing on ”eating disorders” and ”immigration/acculturation”, as well as ”the distinct psychological profiles in the eating psychopathology, general psychopathology, and personality traits”. At the end of the research, the authors mention ”the need for individualized interventions that consider sociocultural contexts”, meaning that these contexts were also considered.

Apart from this mixed research content, the study group has some weak points:

  • There is a big difference between the two compared groups (nationals and migrants) as well as for women/men. Please explain.
  •  Inclusion/exclusion criteria must be carefully chosen, due to the large number of variables considered for the research
  • There is a single-center study, and the characteristics of the region/population must be considered, so the results cannot be generalized 
  • The authors also consider for the present research the Cognitive therapy treatment and 3 meals /day with ” included education on problem-solving strategies, cognitive restructuring, emotional regulation, and improving self-esteem and body image, as well as relapse prevention. Additionally, the therapy addressed eating-related symptomatology through psychoeducation, dietary monitoring, and normalization of nutritional patterns ”

  • Cronbach's alpha for the EDI-2 ranged from 0.605 (asceticism) - please explain

Even if the database is generous, I think that, in this version, is a mixture of data that must be simplified and find a red line that units all the variables of the research.

Author Response

We thank the reviewer for the valuable comments, as well as for recognizing the relevance and potential of our study. We agree that the initial version of the manuscript presented a wide range of variables, hypotheses, and outcomes, which may have given the impression of dispersion. In response to this concern, we have substantially simplified and clarified the focus of the manuscript. Specifically, we reorganized the entire paper around two overarching domains: clinical presentation and treatment response. This dichotomous structure more accurately reflects the logic of our analyses and unifies the different variables assessed (eating psychopathology, general psychopathology, personality traits, and treatment outcomes) without listing them separately or fragmenting the narrative. We revised the Abstract, Introduction, Results, and Discussion to ensure consistency with this clearer conceptual framework, and we believe that this restructuring now provides a coherent “red line” connecting all aspects of the research.

Specific responses to the reviewer’s additional comments are provided below.

Thank you for the opportunity to review the research article entitled ”Eating disorders in an immigrant population: are clinical features and treatment outcomes different from the native-born Spanish population?” It is a challenging research project, well-written, with potential for publication. 

In my opinion, the manuscript should be simplified and the research must be clarified, due to the many goals, variables, and hypotheses that the authors are presenting, even from the beginning: the research teams specified that there are results related to ... and please check the Abstract and also the manuscript:

”drive for thinness and body dissatisfaction .... interpersonal distrust, maturity fears, perfectionism, anxiety symptoms, and self-transcendence. Also, they present data about  ”dropout rates, poorer clinical outcomes, and lower remission...” Simultaneously, the title is focusing on ”eating disorders” and ”immigration/acculturation”, as well as ”the distinct psychological profiles in the eating psychopathology, general psychopathology, and personality traits”. At the end of the research, the authors mention ”the need for individualized interventions that consider sociocultural contexts”, meaning that these contexts were also considered.

Response. The reviewer is certainly correct in noting that the initial formulation of the study aims and hypotheses included too many elements, which created an impression of dispersion. In response we have simplified and clarified the focus of the manuscript. Specifically, we now refer to the two core domains examined: (1) clinical presentation and (2) treatment response. This wording more accurately reflects the structure of our analyses and improves the conceptual coherence of the paper.

By using these two categories, we integrate the different variables assessed (eating psychopathology, general psychopathology, personality traits, and treatment outcomes) without listing them separately, avoiding redundancy and reducing complexity.

We revised the Abstract, Introduction, and Discussion to ensure that the aims, results, and implications are presented consistently within this simplified framework.

Apart from this mixed research content, the study group has some weak points:

- There is a big difference between the two compared groups (nationals and migrants) as well as for women/men. Please explain.

Response. We addressed the issue in the limitations paragraph: “Sixth, there was a marked imbalance between native-born and immigrant patients, as well as between women and men. These discrepancies reflect the actual distribution of patients in our clinical setting, but they nonetheless limit generalizability and reduce statistical power for subgroup analyses.”

- Inclusion/exclusion criteria must be carefully chosen, due to the large number of variables considered for the research

Response. We appreciate the reviewer’s comment and understand that the observation refers to the set of variables included and excluded in the present study. In this regard, we would like to clarify that the study has an exploratory nature; therefore, it was considered appropriate to include a relatively broad range of constructs and variables in order to obtain a more comprehensive understanding of the phenomenon under investigation. However, although the overall number of variables is high, the number of dimensions analyzed is not equally large: in addition to therapeutic outcomes, the study includes measures in four main areas: sociodemographic variables, eating disorder symptomatology, general psychopathology, and personality traits. It is also important to note that, for the clinical constructs, we used well-validated instruments that contain a relatively high number of scales (EDI-2 for measuring eating related problems, SCL-90-R for psychopathology, and TCI-R for personality). The analysis of these domains in detail is one of the strengths of the study, as it allows us to capture clinical nuances that would not be detectable with more restrictive approaches or with instruments covering fewer dimensions.

We hope this clarification helps to contextualize our decisions regarding the inclusion/exclusion criteria and the breadth of the variables considered.

- There is a single-center study, and the characteristics of the region/population must be considered, so the results cannot be generalized

Response. We addressed the issue in the limitations paragraph: “Finally, this was a single-center study, and the characteristics of the region and the population treated in our unit must be considered. Consequently, the findings cannot be generalized beyond similar clinical settings.”

- The authors also consider for the present research the Cognitive therapy treatment and 3 meals /day with ” included education on problem-solving strategies, cognitive restructuring, emotional regulation, and improving self-esteem and body image, as well as relapse prevention. Additionally, the therapy addressed eating-related symptomatology through psychoeducation, dietary monitoring, and normalization of nutritional patterns ”

- Cronbach's alpha for the EDI-2 ranged from 0.605 (asceticism) - please explain

Response. Thank you for your observation. The value reported in the manuscript was a typo. The correct Cronbach’s alpha for the asceticism subscale of the EDI-2 is 0.718, not 0.605 as mistakenly indicated. Although this value may appear somewhat low (as it falls below 0.80), the scientific literature generally considers internal consistency coefficients above 0.70 acceptable when they are calculated in the specific samples of each study. We have corrected this in the revised version of the manuscript (Taber, 2018).

Taber, K.S. (2018) The Use of Cronbach’s Alpha When Developing and Reporting Research Instruments in Science Education. Research in Science Education, 48, 1273–1296. https://doi.org/10.1007/s11165-016-9602-2

Even if the database is generous, I think that, in this version, is a mixture of data that must be simplified and find a red line that units all the variables of the research.

Response. We hope that the clearer two-part structure (distinguishing more explicitly between clinical features and treatment outcomes) now provides a coherent through-line for the manuscript and helps unify the different variables examined in the study.

Reviewer 2 Report

Comments and Suggestions for Authors
  1. Significance of the Topic and Overall Assessment

The article addresses an important and timely topic concerning cultural differences in the clinical presentation of eating disorders (EDs) and in treatment outcomes. Given the increasing migration in Europe and the scarcity of research on the effectiveness of ED treatments in immigrant populations, the study represents a valuable contribution. The authors use a large clinical sample (N = 1104), which enhances the statistical reliability of their analyses.

At the same time, the article exhibits several significant methodological, interpretative, and conceptual gaps that limit the strength of the conclusions and require critical discussion.

  1. Strengths of the Study

Large clinical sample
The sample includes 1104 patients (947 native-born and 157 immigrants), making the study one of the largest examining EDs in migrant populations.

High level of tool standardization
The authors employed three widely used and validated instruments:
– EDI-2,
– SCL-90-R,
– TCI-R.
Cronbach’s alpha values reported in Table 2 (p. 6) confirm their adequacy in the analyzed sample.

Analyses adjusted for ED subtype
Treatment outcomes, dropout rates, and regression models were controlled for ED subtype, which is methodologically sound and necessary.

Comprehensive statistical analyses
The authors used ANOVA, Cox regression, and logistic regression with the Hosmer–Lemeshow test. The results are reported clearly (e.g., Tables 2–4).

  1. Critical Methodological Analysis

Missing key information on the migration process

The article does not account for variables such as:
– motivation for migration,
– legal status,
– pre-migration and post-migration trauma,
– migrant generation,
– level of acculturation,
– language barriers in therapy.

These are fundamental moderators of ED risk and treatment outcomes. Although the authors acknowledge this in the Limitations section, the scope of omission significantly affects the interpretation of their findings.

Inconsistency in the rationale for clinical differences

Results indicate that immigrant patients present lower Drive for Thinness and Body Dissatisfaction, but higher anxiety, perfectionism, and interpersonal distrust (Table 2, p. 6).
The authors suggest these features arise from cultural conflict—yet the absence of acculturation data makes such a conclusion difficult to support.

There are no objective measures of cultural distance, rendering the interpretations largely speculative.

Differences in ED subtype distribution are insufficiently analyzed

Table 1 (p. 5) shows statistically significant differences in the distribution of AN, BN, BED, and OSFED across groups.
While the authors adjust for ED subtype in their models, they do not analyze why these subtype distributions differ nor discuss their potential cultural implications.

Lack of analysis of culturally adapted treatment

Immigrant patients received the same CBT protocol developed for the native population.
There is no analysis of:
– language barriers,
– therapeutic adaptation,
– cultural translation of CBT.

Thus, concluding that immigrants have “poorer treatment outcomes” is risky—treatment inefficacy may stem from a lack of adaptation rather than clinical differences.

Problematic interpretation of TCI-R personality traits

Findings show, among immigrant patients:
– lower Reward Dependence,
– higher Persistence,
– higher Self-Transcendence (Table 2, p. 6).

The authors’ interpretation is speculative and does not consider:
– cultural differences in TCI response patterns,
– population norms,
– measurement bias (lack of norms for immigrant groups).

TCI-R is not culturally neutral, yet this issue is not discussed.

4. Discussion – Critical Assessment of Interpretation

The discussion is speculative and at times narrative.
The authors connect psychopathological results with theories of acculturative stress, but do not measure acculturation, stress, or cultural conflict.
Interpretations concerning interpersonal distrust, perfectionism, and maturity fears are psychologically interesting but lack empirical grounding in the presented data.

5. Limitations – Assessment of Adequacy

Although the authors list key limitations, one crucial issue is missing:

Lack of control for linguistic and communicative differences in therapy.

In ED treatment—where language and emotional expression are central—language barriers have major implications. Their omission is a substantial oversight.

Author Response

We thank the reviewer for the constructive evaluation of our manuscript. We appreciate the recognition of the study’s strengths. At the same time, we acknowledge the important methodological and interpretative issues raised. In response, we have substantially revised the manuscript to address these concerns. We clarified the study’s focus and reorganized the presentation of results within a more streamlined framework. We expanded the Limitations section to include several key issues highlighted by the reviewer.We also tempered speculative interpretations in the Discussion and explicitly acknowledged that interpretations related to acculturative stress remain exploratory due to the absence of direct measures.

Further details and specific modifications are provided in the point-by-point responses below.

  1. Significance of the Topic and Overall Assessment

The article addresses an important and timely topic concerning cultural differences in the clinical presentation of eating disorders (EDs) and in treatment outcomes. Given the increasing migration in Europe and the scarcity of research on the effectiveness of ED treatments in immigrant populations, the study represents a valuable contribution. The authors use a large clinical sample (N = 1104), which enhances the statistical reliability of their analyses.

At the same time, the article exhibits several significant methodological, interpretative, and conceptual gaps that limit the strength of the conclusions and require critical discussion.

  1. Strengths of the Study

Large clinical sample
The sample includes 1104 patients (947 native-born and 157 immigrants), making the study one of the largest examining EDs in migrant populations.

High level of tool standardization
The authors employed three widely used and validated instruments:
– EDI-2,
– SCL-90-R,
– TCI-R.
Cronbach’s alpha values reported in Table 2 (p. 6) confirm their adequacy in the analyzed sample.

Analyses adjusted for ED subtype
Treatment outcomes, dropout rates, and regression models were controlled for ED subtype, which is methodologically sound and necessary.

Comprehensive statistical analyses
The authors used ANOVA, Cox regression, and logistic regression with the Hosmer–Lemeshow test. The results are reported clearly (e.g., Tables 2–4).

  1. Critical Methodological Analysis

Missing key information on the migration process

The article does not account for variables such as:
– motivation for migration,
– legal status,
– pre-migration and post-migration trauma,
– migrant generation,
– level of acculturation,
– language barriers in therapy.

These are fundamental moderators of ED risk and treatment outcomes. Although the authors acknowledge this in the Limitations section, the scope of omission significantly affects the interpretation of their findings.

Response. We acknowledge that the absence of migration-specific variables (e.g., motivation for migration, legal status, pre- and post-migration trauma, migrant generation, acculturation level, and language-related barriers) remains a limitation of the present study. Although we were able to include socioeconomic status (which is an important contextual factor) this cannot compensate for the lack of detailed information on the migration process, which would have allowed for a more fine-grained interpretation of the observed group differences. We have addressed these limitations in greater detail in the Limitations section of the Discussion.

Inconsistency in the rationale for clinical differences

Results indicate that immigrant patients present lower Drive for Thinness and Body Dissatisfaction, but higher anxiety, perfectionism, and interpersonal distrust (Table 2, p. 6).
The authors suggest these features arise from cultural conflict—yet the absence of acculturation data makes such a conclusion difficult to support.

There are no objective measures of cultural distance, rendering the interpretations largely speculative.

Response. The reviewer is certainly correct in noting that, without measuring acculturation, our interpretations cannot go beyond speculative hypotheses. We have explicitly acknowledged this limitation in the revised manuscript. Nonetheless, these hypotheses are grounded in prior literature suggesting that cultural conflict and acculturative stress may influence ED-related psychopathology. Importantly, we have tempered and clarified these interpretations in the Discussion to avoid overstating conclusions that cannot be empirically verified with the current dataset.

Differences in ED subtype distribution are insufficiently analyzed

Table 1 (p. 5) shows statistically significant differences in the distribution of AN, BN, BED, and OSFED across groups.
While the authors adjust for ED subtype in their models, they do not analyze why these subtype distributions differ nor discuss their potential cultural implications.

Response. We added the following statement at the beginning of the discussion: “We observed significant differences in the distribution of ED subtypes between the two groups. These differences likely reflect real patterns within our clinical catchment area, where cultural norms, help-seeking behaviors, symptom expression, and access to specialized care vary across populations; however, because we did not collect migration-specific or cultural variables, we cannot determine the precise cultural or contextual factors driving these subtype disparities.”

Lack of analysis of culturally adapted treatment

Immigrant patients received the same CBT protocol developed for the native population.
There is no analysis of:
– language barriers,
– therapeutic adaptation,
– cultural translation of CBT.

Thus, concluding that immigrants have “poorer treatment outcomes” is risky—treatment inefficacy may stem from a lack of adaptation rather than clinical differences.

Response. Thank you for this important comment. We agree that language proficiency, therapeutic adaptation, and cultural translation are relevant aspects when interpreting differences in treatment outcomes. We would like to clarify that in this study all patients, both immigrants and non-immigrants, were referred to and treated in the same specialized tertiary unit of the public health system. This unit serves the entire population of a single, well-defined geographic area, which ensures homogeneous referral criteria and comparable access pathways for all individuals included in the study. In addition, to minimize potential communication barriers, all participants were required to be proficient in both official languages of the region where the hospital is located, thus reducing the likelihood that basic linguistic difficulties could have interfered with treatment delivery.

We also compared immigrant and non-immigrant participants on key sociodemographic variables such as sex, age, educational level, and social position index, and no significant differences were found between groups. Similarly, there were no differences in age of onset or duration of the disorder, suggesting broadly comparable clinical trajectories prior to treatment. Regarding the intervention, immigrant and non-immigrant patients received the same CBT protocol, which constitutes the standard treatment in this specialized unit for eating disorders. Although this protocol was originally developed for the native population, its application followed routine clinical practice within the public health system.

In light of these considerations, we believe that the potential impact of language barriers or lack of cultural adaptation is substantially reduced, although we acknowledge that such factors may still play a role and merit further investigation. In this sense, and to address the reviewer’s concern, we would like to emphasize that one of the aims of our study is precisely to gather the empirical evidence needed to better understand the specific needs of immigrant patients and, ultimately, to inform future adaptations of standard treatments when appropriate. We have incorporated this clarification into the revised manuscript and adjusted the interpretation of group differences in treatment outcomes accordingly.

Problematic interpretation of TCI-R personality traits

Findings show, among immigrant patients:
– lower Reward Dependence,
– higher Persistence,
– higher Self-Transcendence (Table 2, p. 6).

The authors’ interpretation is speculative and does not consider:
– cultural differences in TCI response patterns,
– population norms,
– measurement bias (lack of norms for immigrant groups).

TCI-R is not culturally neutral, yet this issue is not discussed.

Response. We appreciate the reviewer’s important observation regarding the cultural interpretation of TCI-R scores. It is true that the TCI-R is not culturally neutral, and that population-specific norms for diverse immigrant groups are lacking. For this reason, we avoided drawing strong cultural conclusions from the personality findings and framed our interpretations cautiously as hypothetical, in line with existing literature.

Moreover, our primary aim was comparative: both groups completed the same validated Spanish version of the TCI-R under identical clinical conditions. This allows for internal comparison within our sample, even though broader cross-cultural generalization is limited. We now explicitly acknowledge in the Limitations section that cultural response styles may influence TCI-R profiles and the lack of immigrant-specific normative data restricts interpretability (“Likewise, we did not evaluate potential cultural variation in personality constructs. Since personality measures such as the TCI-R are not culturally neutral, the lack of popula-tion-specific norms for immigrant patients represents an additional source of measure-ment bias that must be acknowledged.”)

  1. Discussion – Critical Assessment of Interpretation

The discussion is speculative and at times narrative.
The authors connect psychopathological results with theories of acculturative stress, but do not measure acculturation, stress, or cultural conflict.
Interpretations concerning interpersonal distrust, perfectionism, and maturity fears are psychologically interesting but lack empirical grounding in the presented data.

Response. We agree with the reviewer that, without direct measures of acculturation, migration-related stress, or cultural conflict, interpretations linking our findings to acculturative stress models must remain tentative. To address this concern, we revised the Discussion by explicitly stating that these interpretations are exploratory. In the opening paragraph of the Discussion, we now clarify that although some psychopathological patterns are framed within theoretical models of acculturative stress, we did not assess acculturation, cultural conflict, or migration-related stress, and therefore such interpretations should be treated with caution.

  1. Limitations – Assessment of Adequacy

Although the authors list key limitations, one crucial issue is missing:

Lack of control for linguistic and communicative differences in therapy.

In ED treatment—where language and emotional expression are central—language barriers have major implications. Their omission is a substantial oversight.

Response. We agree with the Reviewer. We addressed the issue in the limitations paragraph: “Third, we did not systematically assess linguistic or communicative differences. Although patients admitted to treatment were required to have sufficient language comprehension to participate, we did not quantify language proficiency nor examine whether commu-nication style differences, subtle language barriers, or culturally shaped emotional ex-pression may have influenced treatment engagement or outcomes. Given that ED treatment relies heavily on verbal processing and emotional exploration, this omission is an important limitation.”

Reviewer 3 Report

Comments and Suggestions for Authors

After careful consideration, I believe that the manuscript entitled “Eating disorders in an immigrant population: are clinical features and treatment outcomes different from the native-born Spanish population?” is not suitable for publication in its current form. The main issue concerns the statistical analysis.

Here are my comments:

1. Section 2.3.2: The authors reported Cronbach’s alpha values for the questionnaires assessed; however, they did not describe what Cronbach’s alpha measures (internal consistency) or what minimum values are considered indicative of good consistency. This information should be included in the manuscript.

2. Statistical Analysis section

2.1. The authors stated that ANOVA was used to compare groups for quantitative variables. However, the appropriate test is the Student’s t-test. Please update this information in the manuscript.

2.2. The results of the Cox regression model are not clear, and Figure 4 is not consistent with a Cox regression model. In fact, it appears that Figure 4 does not represent the results of a Cox model but rather Kaplan–Meier estimates. Furthermore, it is not clear why an adjustment for ED subtype is necessary in the dropout analysis. Therefore, my suggestion is to replace the Cox regression with Kaplan–Meier curves and compare immigrants and non-immigrants using the Log-rank test.

2.3. The authors indicate that Finner’s procedure was adopted to avoid Type I error due to multiple testing. However, it is not clear in the manuscript how this correction was applied. How many multiple tests were corrected using this procedure? Please indicate in each table whether the p-values were corrected using Finner’s procedure (and, in each case, how many multiple tests the correction considered).

2.4. Include in this section information regarding the effect sizes presented (Cramer–von Mises and Cohen’s d) and specify the thresholds (i.e., which intervals correspond to small, medium, or large effect sizes).

2.5. The logistic regression goodness-of-fit considered the Hosmer–Lemeshow test and Nagelkerke’s R². Please indicate in the manuscript from which R² value the logistic regression model is considered to have a good fit.

3. Tables 1–3. Describe the effect sizes (“C-V” and “d”) in the footnotes. See comment 2.4.

4. Table 2: ANOVA must be replaced by the Student’s t-test. See comment 2.1.

5. Figure 2: It is not clear what the percentages in the figure represent. What does “Percentage of patients scoring within the clinical range across major psychopathology domains” mean? Please include an explanation in the manuscript clarifying what this percentage represents. Moreover, the line chart is inappropriate in this context, as it gives a misleading impression of continuity between items. A bar chart would be more suitable.

6. Table 2. The significant chi-square test (p = 0.012) should be followed by a post hoc analysis. Since four outcomes are being compared, it is necessary to identify where the difference lies. For example, is there a difference between “Partial remission” and “Full remission”? The same applies to the analysis adjusted for ED subtype (logistic regression): it is necessary to identify where the difference is found. Additionally, how were the percentages adjusted for ED subtype obtained? Were they produced by the logistic regression?

7. Table 4: What is considered an adequate value of “NR²”? See comment 2.5.

8. Figure 4: This figure does not appear to present the results of the Cox regression model. See comment 2.2.

9. All tables and figures: The authors alternate between the terms “Native-born,” “Immigrant (-),” and “Immigrant (no).” Please standardize these terms throughout the manuscript. The same applies to “Immigrant,” “Immigrant (+),” and “Immigrant (yes).”

Author Response

We thank the reviewer for the rigorous evaluation, which led to substantial methodological and editorial improvements. We clarified the statistical framework, updated terminology, improved the transparency of analytic procedures, corrected and expanded explanations of reliability, effect sizes, and model fit, and revised figures and tables for accuracy and consistency. We also incorporated the reviewer’s suggestions regarding survival analysis, post-hoc comparisons, and standardization of labels.

Specific responses to the reviewer’s additional comments are provided below.

After careful consideration, I believe that the manuscript entitled “Eating disorders in an immigrant population: are clinical features and treatment outcomes different from the native-born Spanish population?” is not suitable for publication in its current form. The main issue concerns the statistical analysis.

Here are my comments:

  1. Section 2.3.2: The authors reported Cronbach’s alpha values for the questionnaires assessed; however, they did not describe what Cronbach’s alpha measures (internal consistency) or what minimum values are considered indicative of good consistency. This information should be included in the manuscript.

Response. Thank you for the valuable input. We have added this information at the beginning of Section 2.3.2 Self-Report Measures.

  1. Statistical Analysis section

2.1. The authors stated that ANOVA was used to compare groups for quantitative variables. However, the appropriate test is the Student’s t-test. Please update this information in the manuscript.

Response. Thank you for this observation. We would like to clarify that the Student’s T-test is indeed a particular case of the ANOVA model when comparing the means of two groups. Both procedures provide exactly the same p-value, as the F statistic obtained in the ANOVA is mathematically equivalent to the square of the t statistic. In this sense, ANOVA can be regarded as a general framework that includes the t-test as a special case. We used the ANOVA terminology because it is the broader model on which the comparison was based, and because it is the usual procedure used in our research team because it allows the inclusion of control variables (covariates) when needed. Nonetheless, we appreciate the reviewer’s suggestion, and, for the sake of clarity, we have updated the manuscript to specify that the comparison between two groups is equivalent to a Student’s T-test.

2.2. The results of the Cox regression model are not clear, and Figure 4 is not consistent with a Cox regression model. In fact, it appears that Figure 4 does not represent the results of a Cox model but rather Kaplan–Meier estimates. Furthermore, it is not clear why an adjustment for ED subtype is necessary in the dropout analysis. Therefore, my suggestion is to replace the Cox regression with Kaplan–Meier curves and compare immigrants and non-immigrants using the Log-rank test.

Response. Thank you for this comment. We would like to clarify that a Cox regression model was used because the survival curves were adjusted for diagnostic subtype, as stated in the manuscript. The analytic approach was chosen to obtain adjusted survival estimates and an adjusted comparison between immigrants and non-immigrants. This adjustment is not possible when survival curves are estimated using the Kaplan–Meier method and compared with the Log-rank test, as these procedures do not allow the inclusion of covariates. We have revised the manuscript to clarify that Figure 4 represents adjusted survival functions derived from the Cox regression model.

The SPSS procedure we used generates adjusted survival curves derived from the Cox model and provides the corresponding group comparison within the same regression framework. For transparency, we now include the specific SPSS command employed to obtain the adjusted curves and the associated statistics:

COXREG        dropoutTime

  /STATUS=    dropout(1)

  /METHOD=  ENTER Inmigrant DSM_5type

  /CONTRAST (DSM_5type)=Indicator

  /PRINT=CI(95)

  /PLOT SURVIVAL

  /CRITERIA=PIN(.05) POUT(.10) ITERATE(20).

2.3. The authors indicate that Finner’s procedure was adopted to avoid Type I error due to multiple testing. However, it is not clear in the manuscript how this correction was applied. How many multiple tests were corrected using this procedure? Please indicate in each table whether the p-values were corrected using Finner’s procedure (and, in each case, how many multiple tests the correction considered).

Response. Thank you for this comment. We have now clarified how Finner’s procedure was applied in the manuscript. Specifically, we applied Finner’s procedure to the full set of quantitative and categorical group comparisons contained in tables 1 and 2. We have updated both tables to indicate in the table footnotes that the reported p-values correspond to the Finner-adjusted results, to enhance transparency and reproducibility.

2.4. Include in this section information regarding the effect sizes presented (Cramer–von Mises and Cohen’s d) and specify the thresholds (i.e., which intervals correspond to small, medium, or large effect sizes).

We did it reporting the following paragraph “Effect sizes were reported using Cramer’s V (for categorical comparisons) and Cohen’s d (for quantitative comparisons). For Cramer’s V, values of approximately 0.10, 0.30, and 0.50 were considered small, medium, and large effects, respectively. For Cohen’s d, values around 0.20, 0.50, and 0.80 were interpreted as small, medium, and large effect sizes.” in section 2.5.

2.5. The logistic regression goodness-of-fit considered the Hosmer–Lemeshow test and Nagelkerke’s R². Please indicate in the manuscript from which R² value the logistic regression model is considered to have a good fit.

Response. Thank you for this comment. We have clarified in the revised manuscript that the goodness of fit of the logistic regression models was assessed using the Hosmer–Lemeshow test. Values of p > 0.05 in this test indicate that the model fits the data adequately.

Regarding Nagelkerke’s pseudo-R², we note that this statistic reflects the global predictive capacity of the model but does not have an established threshold that defines when a model should be considered to have a “good” fit (Hughes et al., 2019). Its magnitude depends strongly on the number and nature of the predictors included in the model. Models with fewer variables typically yield lower pseudo-R² values, whereas models with a greater number of predictors tend to show higher predictive capacity by definition.

Although Nagelkerke’s R² should always be interpreted contextually rather than against strict cut-off values, the ranges commonly reported in applied behavioral research can offer a general orientation (Nagelkerke, 1991): values around 0.02–0.05 are typically considered (quasi)null effects, values between 0.05–0.10 are interpreted as poor, values in the 0.10–0.20 range are interpreted as acceptable or moderate fit, values between 0.20–0.30 indicate good fit, and values above 0.30 suggest very good fit, although these higher values tend to be uncommon in the social and behavioral sciences.

In line with this, Model 2 in Table 4 shows a higher Nagelkerke’s R² than Model 1, despite both models including the same number of predictors, which indicates better overall predictive performance in Model 2.

We have incorporated this explanation into the revised manuscript to clarify the interpretation of these goodness-of-fit indices.

References:

Hughes, G., Choudhury, R. A., & McRoberts, N. (2019). Summary Measures of Predictive Power Associated with Logistic Regression Models of Disease Risk. Phytopathology109(5), 712–715. https://doi.org/10.1094/PHYTO-09-18-0356-LE

Nagelkerke, N. J. D. (1991). A note on a general definition of the coefficient of determination. Biometrika, 78(3), 691-692. https://doi.org/10.1093/biomet/78.3.691

  1. Tables 1–3. Describe the effect sizes (“C-V” and “d”) in the footnotes. See comment 2.4.

Thank you for the observation. We have now described the effect sizes (“C-V” and “d”) directly in the footnotes of Tables 1–3, as requested.

  1. Table 2: ANOVA must be replaced by the Student’s t-test. See comment 2.1.

Response. Thank you for this remark. As indicated in our response to Comment 2.1, we have clarified in the manuscript that the comparison between the two groups is equivalent to a Student’s t-test. This correction has already been implemented in Table 2, where the corresponding terminology has been updated accordingly.

  1. Figure 2: It is not clear what the percentages in the figure represent. What does “Percentage of patients scoring within the clinical range across major psychopathology domains” mean? Please include an explanation in the manuscript clarifying what this percentage represents. Moreover, the line chart is inappropriate in this context, as it gives a misleading impression of continuity between items. A bar chart would be more suitable.

Response. Thank you for this comment. We are aware that the scores obtained for the SCL-90-R scales do not constitute a dimensional measure. However, we initially chose to use a line chart because users of this questionnaire commonly employ this format to graphically represent the symptomatic profiles of individuals (as indicated in the instrument’s manual). Since both clinicians and researchers are accustomed to this type of representation, it seemed preferable to maintain this format in order to facilitate interpretation. However, in response to the Reviewer’s comment, we have decided to change the line-plot for a bar-chart. In addition, we have provided an explanation of what the percentages represent, both in the main text and in a footnote accompanying the figure. Specifically, normative scales allow raw scores to be converted into standardized scores (percentiles and T-scores), and established cut-off points indicate when scores deviate from population averages, suggesting an increased risk of psychopathology. For the SCL-90-R, T-scores above 63 are considered significantly elevated and indicate the presence of symptoms with potential clinical relevance.

  1. Table 2. The significant chi-square test (p = 0.012) should be followed by a post hoc analysis. Since four outcomes are being compared, it is necessary to identify where the difference lies. For example, is there a difference between “Partial remission” and “Full remission”? The same applies to the analysis adjusted for ED subtype (logistic regression): it is necessary to identify where the difference is found. Additionally, how were the percentages adjusted for ED subtype obtained? Were they produced by the logistic regression?

Response. Thank you for this comment. We understand that the reviewer is referring to the results presented in Table 3. We have now performed the post-hoc contrasts as suggested and have included in the text the specific groups that differ:  

“The risk of treatment outcomes is displayed in Table 3 and Figure 3. Post-hoc comparisons for the outcome analyzed in four groups (dropout, non remission, partial remission and full remission) showed that compared to native-born patients, immi-grant patients were more likely to dropout (p = .003). Considering the binary outcome poor versus good outcome, immigrant patients also achieved higher likelihood of poorer results. These differences remained significant even after adjusting for ED sub-types (the contrast for comparing the likelihood of dropout between native-born and immigrant patients achieved significant result p = .014)”.

Regarding the percentages adjusted for ED subtype, these values were obtained from the logistic regression model, as predicted probabilities for each group. For example, this is the syntax for obtaining the prevalences for the value=0 of the CBToutcome (dropout in our study) for each group:

TEMPORARY.

RECODE outcome (0=1) (1=0) (2=0) (3=0).

LOGISTIC REGRESSION VARIABLES outcome

  /METHOD=ENTER immigrant DSM_5type 

  /CONTRAST (DSM_5type  )=Indicator

  /PRINT=GOODFIT CI(95) /SAVE PRED

  /CRITERIA=PIN(0.90) POUT(0.95) ITERATE(20) CUT(0.5).

MEANS PRE_1 BY immigrant.

DELETE VARIABLES PRE_1 .

  1. Table 4: What is considered an adequate value of “NR²”? See comment 2.5.

Response. Thank you for this remark. We have already addressed this question in response to Comment 2.5, where we clarified the interpretation of Nagelkerke’s R² and its dependence on the number of predictors included in the model.

  1. Figure 4: This figure does not appear to present the results of the Cox regression model. See comment 2.2.

Response. Thank you for this comment. We have already addressed this point in response to Comment 2.2, where we clarified that Figure 4 presents survival curves adjusted for ED subtype obtained from the Cox regression model and explained the SPSS procedure used to generate the figure.

  1. All tables and figures: The authors alternate between the terms “Native-born,” “Immigrant (-),” and “Immigrant (no).” Please standardize these terms throughout the manuscript. The same applies to “Immigrant,” “Immigrant (+),” and “Immigrant (yes).”

We have standardized the terminology across all tables and figures: we consistently use the dichotomy “native-born” and “immigrant.”

Round 2

Reviewer 1 Report

Comments and Suggestions for Authors

I have viewed an improved version of the manuscript but I keep a few comments for a final minor revision:
- although the authors are doing a comparative study between natives and immigrants, putting all migrants in the same category is not very objective because the cultural subtype is very different when it comes to health, body image, eating behavior, etc. If the authors do not want to detail this (and it would be difficult since the migrant group is significantly smaller than the native population investigated) then the researchers should mention these aspects in the discussions or within the study limits.

Comments on the Quality of English Language

I have viewed an improved version of the manuscript but I keep a few comments for a final minor revision:
- although the authors are doing a comparative study between natives and immigrants, putting all migrants in the same category is not very objective because the cultural subtype is very different when it comes to health, body image, eating behavior, etc. If the authors do not want to detail this (and it would be difficult since the migrant group is significantly smaller than the native population investigated) then the researchers should mention these aspects in the discussions or within the study limits.

Author Response

Comments and Suggestions for Authors

I have viewed an improved version of the manuscript but I keep a few comments for a final minor revision:
- although the authors are doing a comparative study between natives and immigrants, putting all migrants in the same category is not very objective because the cultural subtype is very different when it comes to health, body image, eating behavior, etc. If the authors do not want to detail this (and it would be difficult since the migrant group is significantly smaller than the native population investigated) then the researchers should mention these aspects in the discussions or within the study limits.

Response. We thank the Reviewer for highlighting this important point. We agree that grouping all immigrants into a single category risks obscuring cultural heterogeneity relevant to ED presentation and treatment response. Although the small size of the immigrant sample did not allow further stratification, we have now explicitly acknowledged this limitation in the revised manuscript within the second paragraph of the Limitations section and clarified that cultural variability may contribute to unobserved differences between native-born and immigrant patients.

Reviewer 2 Report

Comments and Suggestions for Authors

The authors took most of my comments into account.

Author Response

Thank you!

Reviewer 3 Report

Comments and Suggestions for Authors

I believe the authors have made the corrections and/or satisfactorily addressed the questions. Therefore, I believe the manuscript is now suitable for publication in Nutrients.

Author Response

Thank you!